# Secretion of Interleukin 6 in Human Skeletal Muscle Cultures Depends on Ca^2+^ Signalling

**DOI:** 10.3390/biology12070968

**Published:** 2023-07-07

**Authors:** Blanca Calle-Ciborro, Teresa Espin-Jaime, Francisco J. Santos, Ana Gomez-Martin, Isaac Jardin, Maria J. Pozo, Juan A. Rosado, Pedro J. Camello, Cristina Camello-Almaraz

**Affiliations:** 1Department of Physiology, Instituto de Biomarcadores Patológicos Moleculares y Metabólicos, Universidad de Extremadura, 10003 Cáceres, Spain; bcallecii@unex.es (B.C.-C.); ijp@unex.es (I.J.); mjpozo@unex.es (M.J.P.); jarosado@unex.es (J.A.R.); mcca@unex.es (C.C.-A.); 2Faculty of Medicine, Hospital Universitario, Universidad de Extremadura, 06006 Badajoz, Spain; matespin@unex.es; 3Hospital Universitario, 10004 Cáceres, Spain; cirugiasantos@gmail.com; 4Department of Nursing, Faculty of Nursing and Occupational Therapy, Universidad de Extremadura, 10003 Cáceres, Spain; anagomez@unex.es

**Keywords:** IL-6, Ca^2+^ signals, skeletal muscle, voltage operated calcium channels, store operated calcium channels, intracellular calcium stores

## Abstract

**Simple Summary:**

Skeletal muscle releases numerous hormones into circulation that interact with other organs, such as the liver, bone or the brain. These hormones, termed myokines, mediate the effects of physical activity in health, aging and disease. Interleukin 6 was the first discovered myokine and is actively investigated due to its participation in inflammation, immunity and metabolism. However, there is little information regarding the mechanisms that induce its release from muscle cells, especially in humans. Our aim was to investigate whether changes in the concentration of calcium ions participate in the stimulated release of interleukin 6 in human muscle cells. Using muscle cultures, we have found that several proteins responsible for the calcium increase during stimulation induce the release of interleukin 6 from the muscle cells. This could help to unveil how interleukin 6 and other myokines are released in pathological conditions such as trauma, infections or cancer.

**Abstract:**

The systemic effects of physical activity are mediated by the release of IL-6 and other myokines from contracting muscle. Although the release of IL-6 from muscle has been extensively studied, the information on the cellular mechanisms is fragmentary and scarce, especially regarding the role of Ca^2+^ signals. The aim of this study was to characterize the role of the main components of Ca^2+^ signals in human skeletal muscle cells during IL-6 secretion stimulated by the Ca^2+^ mobilizing agonist ATP. Primary cultures were prepared from surgical samples, fluorescence microscopy was used to evaluate the Ca^2+^ signals and the stimulated release of IL-6 into the medium was determined using ELISA. Intracellular calcium chelator Bapta, low extracellular calcium and the Ca^2+^ channels blocker La^3+^ reduced the ATP-stimulated, but not the basal secretion. Secretion was inhibited by blockers of L-type (nifedipine, verapamil), T-type (NNC55-0396) and Orai1 (Synta66) Ca^2+^ channels and by silencing Orai1 expression. The same effect was achieved with inhibitors of ryanodine receptors (ryanodine, dantrolene) and IP3 receptors (xestospongin C, 2-APB, caffeine). Inhibitors of calmodulin (calmidazolium) and calcineurin (FK506) also decreased secretion. IL-6 transcription in response to ATP was not affected by Bapta or by the T channel blocker. Our results prove that ATP-stimulated IL-6 secretion is mediated at the post-transcriptional level by Ca^2+^ signals, including the mobilization of calcium stores, the activation of store-operated Ca^2+^ entry, and the subsequent activation of voltage-operated Ca^2+^ channels and calmodulin/calcineurin pathways.

## 1. Introduction

The report of IL-6 release by skeletal muscles during contraction to an extent enough to increase its circulating levels [1] opened the way for the experimental characterization of muscle as an endocrine organ [2]. The available evidence shows that skeletal muscle releases a growing list of intercellular messengers collectively termed myokines [3,4]. Because myokines include both local-acting cytokines and canonical hormones involved in a wide range of functions, from metabolism to immunity and inflammation, the field has raised interest due to its translational implications [5,6]. There is experimental evidence that IL-6 and other myokines can account for systemic changes present in inflammatory diseases [7].

IL-6 secretion has been shown to be released directly from muscular cells in cell culture experiments but also in in vivo experiments, where IL-6 has been shown to be released from muscle tissue in response to contraction ([8]; for a review, see [9]). The cellular contraction induced by direct electrical stimulation releases IL-6 both in rodent [10,11] and human cultures [12,13]. This release is considered the main mechanism for the exercise-induced increase in IL-6 during physical activity [1,9]. Direct evidence shows that IL-6 is released from cell fibres during contraction [14].

In spite of the abundant literature on muscle release of IL-6 in the last two decades, evidence regarding the underlying cellular and molecular mechanism is scarce. Culture experiments suggest that the initial stimuli leading to secretion seem to be membrane depolarization (see above), mechanical stress [15] and purinergic receptors [11,16]. A series of reports in rodent cultures and adult fibres have shown that sarcolemma depolarization releases through pannexin channels sufficient ATP to activate purinergic receptors leading to IL-6 release [11,17,18]. Other cellular stimuli could also induce IL-6 release. Adrenergic receptors can induce secretion in rodent muscles [19,20] and cultures [20] and also in humans during exercise [21]. Nitric oxide [22] and hyperthermia [23] can also release IL-6 in rodent models.

Concerning the intracellular pathways involved in IL-6 secretion, reports in human and rodent models have identified the involvement of several transduction pathways, such as Stat3 [11], JNK/MAPK [20], Srf [24] and AMPK [25,26]. In addition, the transcription factors NF-κB and AP-1 [18] and the epigenetic regulator HDAC5 [26] seem to promote and inhibit IL-6 release. The transcription and release of IL-6 in muscle seem to involve some feedback mechanisms similar to other tissues. It has been proposed that IL-6 exerts a positive autocrine loop in rodent muscle cells involving mRNA synthesis [11]. In line with this, it has been proposed that the IL-6 secretion is regulated at the level of mRNA processing through regulatory proteins interfering with transcription [9,27].

A few groups have also reported the participation of some elements of the Ca^2+^ signals in IL-6 secretion. Current evidence indicates that both IP_3_R [11] and RyR [28] Ca^2+^ stores are involved. Extracellular Ca^2+^ ions [29] and cytosolic Ca^2+^ signals [16,25] seem also to be required in rodent cultures, but the Ca^2+^ influx pathways have not been studied. The participation of Ca^2+^ independently of signalling mechanisms linked to receptors is also supported by the finding that calmodulin [30] in mouse lymphocytes and calcineurin in human skeletal cultures [31] and rodent muscle and cultures [32] can mediate IL-6 release.

Given that the available information on this aspect is fragmentary and almost inexistent for human muscle, the objective of this study was to fully characterize the role of the main components of the Ca^2+^ signal in the IL-6 release in human skeletal cells in culture.

## 2. Materials and Methods

### 2.1. Isolation and Culturing of Human Primary Skeletal Muscle Cells

The study was approved by the Ethical Committees of the University of Extremadura and Hospital Universitario de Extremadura (Servicio Extremeño de Salud) (ref IB18025) and followed the rules of the Declaration of Helsinki of 1975 (https://www.wma.net/what-we-do/medical-ethics/declaration-of-helsinki/, accesed on 31 March 2023), revised in 2013. Human primary cultures were prepared from small fragments of rectus abdomini muscle discarded during laparotomy, collected in a sterile Krebs–Henseleit solution (KH), and cultured on the same day following previously reported methods [33]. The study included cultures from 10 donors, aged 24–70 years: 7 males and 3 females. They were not diagnosed with diabetes, metabolic syndrome, muscle or neurological disorders or hepatic or renal failure, and were not under treatment with anabolic drugs or drugs acting at the muscle endplate.

The digestion of the sample was based on previous reports [34]. Briefly, after the removal of visible blood vessels and fatty tissue, the sample was cut in 1 mm fragments, placed in a 1 mg/mL BSA KH solution and sequentially treated with papain (1 mg/mL, 15 min, 1 mg/mL dithioerythitryol) and collagenase (1 mg/mL, 5 min) at 37 °C. The small fragments were then dispersed with a pipette in DMEM:F12 medium, centrifuged (8 min 1400 rpm) and pre-plated in a Petri dish for 1 h to allow attachment of fibroblasts. The supernatant was then transferred to a collagen-coated culture flask, and the satellite cells were allowed to transform into myoblasts and to proliferate with the growth medium (for composition, see below; 37 °C, 5% CO_2_). The medium was replaced every two days, and cells were replated when approaching confluence (usually after 4–7 days for the original passage) and frozen in liquid nitrogen for experiments at passages 3–5.

For experiments, cells were seeded in 24-well culture plates coated with collagen at 6000–7000 cells/cm^2^ in a growth medium, and at 80–90% confluence, the medium was changed to differentiation medium and replaced every other day. Under these conditions, the proliferating myoblasts fuse into multinuclear myotubes, the in vitro precedent of adult muscle fibres.

The growth medium composition was DMEM:F12 (Hepes and L-glutamine), 20% fetal bovine serum, insulin (10 µg/mL), bFGF (1 ng/mL), EGF (10 ng/mL), dexamethasone (0.4 µg/mL) and antibiotics (penicillin, streptomycin and gentamycin). The differentiation medium was similar but fetal bovine serum was replaced by 2% horse serum, and bFGF, FGF and dexamethasone were omitted.

### 2.2. Secretion Studies

For secretion experiments replicating myoblasts were differentiated during 6–8 days. On the day of the experiment, the cells were washed three times with PBS and placed in 0.5 mL serum-free culture medium (DMEM:F12) to collect the IL-6 secretion. The medium contained no further addition, the stimulus, the appropriated inhibitor or a combination of both. In each culture plate, the secretion was collected under basal (unstimulated) and stimulated conditions both in the absence and presence of the inhibitors assayed. Inhibitors were applied for 15 min before the application of the stimulus (20 min in the case of BAPTA-AM). After a subsequent 2 h incubation, the supernatant was collected, centrifuged and frozen for analysis. To normalize the secretion to the cellular population, nuclei were marked with Hoechst 33342, 30 images/well were captured (EVOS-Fl2 automated microscope) and automatically analysed with a homemade water-shedding macro using Fiji-2 [35].

### 2.3. IL-6 Determination

IL-6 was assayed using MabTech IL-6 ELISA kit (ref. 3460-1HP) following the manufacturer’s instructions.

### 2.4. Ca^2+^ Signal Experiments

Skeletal myotubes were cultured on collagen-coated glass coverslips (0.17 mm thick) following the procedure described above. For experiments, the coverslips were placed in a serum-free culture media containing 4 µM fura 2-AM at room temperature for 45 min, followed by a 20 min period in a physiological Tyrode solution (in mM: 10 HEPES, 137 NaCl, 5.4 KCl, 1.8 CaCl_2_, 1.1 MgCl_2_, 2.2 NaHCO_3_, 0.4 NaHPO_4_ and 5.6 D-glucose, pH 7.3). The coverslip was placed in an experimental chamber mounted on the stage of an inverted microscope (Eclipse TE2000-S; Nikon, Barcelona, Spain) to excite the cells at 340 and 380 nm using a computer-controlled monochromator (Optoscan; Cairn Research, Faversham, UK) at 1 Hz, and the emitted fluorescence images were captured with a CCD camera (ORCAII-ER; Hamamatsu Photonics, Barcelona, Spain) and recorded using dedicated software (Metafluor; Molecular Devices, San Jose, CA, USA). The ratio of fluorescence at 340 nm to 380 nm (F340/F380) was calculated pixel by pixel as an index of cytosolic Ca^2+^ concentration. Once the experimental chamber was placed in the microscope stage, the cells were kept under a constant flow of Tyrode solution, and stimulation was achieved by switching to a solution containing the desired concentration of the stimuli (500 µM ATPγS). When used, inhibitors were applied in the same Tyrode solution for 15 min previous to stimulation. When necessary, a low Ca^2+^ Tyrode was used, substituting 1 mM EGTA for 1.8 CaCl_2_ (0Ca^2+^ Tyrode). To study changes in the Ca^2+^ signals, the peak response was measured as a delta increase in the ratio divided by the previous resting ratio (DF/F0). The integrated or overall response was measured as the area under the curve of the response trace for 180 s after the onset of the Ca^2+^ transient after subtraction of the previous resting value.

### 2.5. Orai 1 Expression Silencing

Cells were transfected with 1 µg/mL shOrai1 or scramble plasmids using Lipofectamine transfection reagent and were used 48 h after transfection. The shOrai1 sense sequence was 5′-CACCTCACTGGTTAGCCATAAGACGAATCTTATGGCTAACCAGTGA-3′, and the antisense sequence was 5′-AAAACCTTTACACGCTAGATGGTTTGCTCTTATGGCTAACCAGTGA- 3′.

### 2.6. Western Blotting

To assess the success of Orai1 silencing, Western blotting assay was used to evaluate the Orai content of the cells. Briefly, after cell lysis (ice-cold Nonidet P-40 buffer pH 8 and complete EDTA-free protease inhibitor tablets), proteins were resolved using 10% SDS-PAGE and electrophoretically transferred onto nitrocellulose membranes for subsequent probing. After blocking residual binding sites (overnight 10% BSA in Tris-buffered saline with 0.1% Tween-20 -TBST-), Orai1 was detected through 1 h incubation with anti-Orai1 antibody (1:1000 in TBST, catalog number O8264, Sigma, Madrid, Spain)) followed by washing (6 × 5 min, TBST) and incubation with goat anti-rabbit IgG conjugated to horseradish peroxidase (1:10,000). For normalization, a primary anti-β-actin antibody was used (1:2000, catalog number: A2066, Sigma, Madrid, Spain). Chemiluminescence was assessed with a ChemidDoc Imaging System (Biorad, Madrid, Spain), and the density of the bands were measured using ImageJ 2.0 software. Data were normalized within each membrane to β-actin.

### 2.7. IL-6 Gene Expression

After RNA extraction with EZNA total RNA kit II (Omega Bio-Tek, Madrid, Spain) and DNA transcription (High-capacity cDNA Reverse Transcription Kit, Thermofisher, Madrid, Spain), the expression of IL-6 gene and GAPDH as constitutive gene for reference were determined using RT-PCR using Taqman probes with accession numbers Hs00174131_m1 for IL-6 and Hs02758991_g1 for GAPDH (ThermoFisher, Madrid, Spain). Expression levels were normalized to GAPDH levels following the 2^−ΔΔCT^ method, using the basal, non-stimulated samples as calibrator [36].

### 2.8. Reagents

DMEM:F12 was obtained from Corning, and fetal bovine serum, horse serum, antibiotics, TNFα, human EGF and human bFGF was obtained from GIBCO. Nifedipine, methoxyverapamil hydrochloride (D-600), lanthanum chloride, Bapta-AM, caffeine, dantrolene, Stattic, Synta66, EGTA, BSA, cycloheximide and collagenase were obtained from Sigma-Merck; papain was obtained from Worthington Biochemicals. Dexamethasone, 2-APB, 5′-N-Ethylcarboxamidoadenosine (NECA), ATPγS, NNC55-0396, 78c (CD38 inhibitor), xestospongin C, FK506 and ryanodine were purchased from Tocris. Insulin and 8BrcADPr were obtained from Santa Cruz. Fura-2 AM was obtained from Invitrogen. All the lipophilic drugs were dissolved in dimethyl sulfoxide (ethanol in the case of xestospongin C). The final concentration of the solvent was ≤0.1%.

### 2.9. Statistics

IL-6 secretion in response to ATPγS or other stimuli was normalized with respect to unstimulated secretion from the same culture plate, either in the absence or in the presence of an inhibitor, as appropriate. Raw IL-6 secretion is expressed as pg/mL × 10^5^ nuclei. Data are given or represented as average ± standard error of the mean (sem). Comparisons were performed using a Student *t*-test (paired when appropriated) or ANOVA followed by planned multiple comparisons, as appropriate. When necessary (lack of normality or heteroscedasticity), the data were transformed (log10 or square root), or a non-parametric test was used.

## 3. Results

Human muscle cultures were differentiated for 1 week and released IL-6 into the culture supernatant in response to several stimuli. Figure 1 shows that both the nonhydrolyzable ATP receptor agonist ATPγS and the adenosine receptor agonist 5′-N-Ethylcarboxamidoadenosine (NECA) induced a dose-dependent and significant increase in the IL-6 concentration in the culture medium. To study the role of Ca^2+^ signalling in IL-6 release, we focused on the response to ATP because it activates Ca^2+^ signals in skeletal muscle cultures and has a key role in responses to membrane depolarization and in the differentiation of muscle cells [11,37].

Intracellular Ca^2+^ signals in many cell types involve an initial and transient increase due to Ca^2+^ release from internal stores followed by a sustained phase of lower magnitude associated with the entry of extracellular Ca^2+^. These components can be selectively studied using established manipulations. A classic approach to block calcium entry is the use of media with low Ca^2+^ concentration (0 Ca^2+^, see composition in Methods Section) or with submillimolar concentration of La^3+^, a generic blocker of calcium channels. Appendix A shows a recording of intracellular Ca^2+^ signals in response to ATP in control and 0 Ca^2+^ conditions. Blockade of calcium entry inhibited the sustained phase of the Ca^2+^ signal but did not modify the initial peak response, reproducing the expected behaviour of calcium-store-based Ca^2+^ signals in the absence of Ca^2+^ entry. Therefore, we compared the control of IL-6 response to ATP with the responses both in the absence of extracellular Ca^2+^ and in the presence of La^3+^. As shown in Figure 2A, ATP-induced response was significantly inhibited under both conditions (*p* < 0.01), indicating that Ca^2+^ influx during the stimulation is necessary for IL-6.

Ca^2+^ signals can also be blocked or strongly reduced by introducing calcium chelators, such as EGTA or BAPTA, into the cells (loaded as acetoxymethyl esters subsequently cleaved by cellular esterases). This can be seen in the cytosolic Ca^2+^ signals of Appendix A. Figure 2B shows that in BAPTA-loaded cells, stimulation with ATP induced strong inhibition in terms of IL-6 release (*p* < 0.009), suggesting that secretion requires increases in cytosolic Ca^2+^. In this series of five experiments using BAPTA-loaded cells, the response to ATP increased IL-6 from 6.47 ± 1.24 (resting) to 11.71 ± 1.67 pg/mL × 10^5^ nuclei, a response statistically smaller (*p* < 0.003, paired *t*-test) than the matched control response, which was from 8.84 ± 1.28 to 72.04 ± 12.37. The finding that the fast Ca^2+^ buffer BAPTA is more effective in preventing ATP-induced IL-6 secretion than the blockade of Ca^2+^ influx strongly suggests that Ca^2+^ release from intracellular stores plays a relevant role in this process.

The effects observed in BAPTA-loaded cells and after the inhibition of Ca^2+^ influx could be due not to the lack of interference with Ca^2+^ signals but only to the unavailability of Ca^2+^ ions in the secretory process of IL-6. If this is true, it is likely that the non-stimulated or basal secretion is equally influenced. However, in our experimental conditions, none of the previous procedures inhibited the basal secretion of IL-6. Figure 3A shows that neither BAPTA, low Ca^2+^ or La^3+^ altered the resting secretion (*p* > 0.6 or greater for any paired comparison, *t*-test). In addition, Figure 3B demonstrates that BAPTA impaired IL-6 release via NECA, an adenosine receptor agonist that potentiates Ca^2+^ signalling in skeletal muscle cells [38], but not by TNFα, which does not produce Ca^2+^ signals in this model.

To further test the role of Ca^2+^ in terms of IL-6 release, we studied the effect of a moderate level of calcium protonophore ionomycin (0.5 µM), a classic method to increase intracellular Ca^2+^. Figure 3A shows that this treatment induced a clear increase in IL-6 secretion, in keeping with a previous report [31]. As a whole, these results indicate that Ca^2+^ signals are involved in the stimulation of IL-6 secretion and that Ca^2+^ ions are not just a required or permissive factor for the release of this cytokine.

There are multiple channels mediating the entry of Ca^2+^ into muscle cells. To assess the participation of voltage-operated Ca^2+^ channels (VOCC) in IL-6 secretion, we used selective inhibitors for the main types present in these cells: L- and T-type VOCCs (Cav1.x and Cav3.x) [39]. Figure 4 shows the effects of nifedipine, the canonical dihydropiridine blocker for Cav1.x, verapamil (D600), another specific Cav1.x inhibitor chemically unrelated, and NNC55-0396, a specific blocker of T-type channels on the Ca^2+^ signal elicited by ATP. As can be seen, the inhibition of the L-type channel mainly inhibits the sustained phase of the signal and, to a lesser extent, the initial peak response. This is consistent with the well-known function of L channels, not only as a Ca^2+^ influx route but as activators of the release from intracellular stores via ryanodine receptor channels.

Figure 5A shows that both nifedipine and verapamil induced a statistically significant inhibition of the secretory ATP response (*p* < 0.005), similar to the effect of NNC55-0396, indicating that Ca^2+^ entry through L-and T-type channels contributes to IL-6 release. Note that in the case of the T channel, there was an almost total suppression of the response, indicating that function of this channel is absolutely required for IL-6 secretion.

We also evaluated the participation of Orai1 Ca^2+^ channels, which are opened due to the depletion of internal Ca^2+^ stores, using the selective inhibitor Synta66 [40] and genetic ablation of the channel (transfecting cells with a silencing plasmid shOrai1). Figure 4B depicts the Ca^2+^ signal evoked by ATP in matched controls, Synta66-treated cells and shOrai silenced cells. The inhibition of capacitative Ca^2+^ entry by any of the two treatments clearly impaired the sustained response. The initial peak response was slightly (non-significantly) reduced in Synta66-treated cells but was clearly inhibited in shOrai1 transfected cells (*p* < 0.005).

In an additional series of experiments (shown in Figure 5B), we determined the IL-6 release in cells treated with Synta66, in cells transfected with shOrai and, for comparison purposes, once again in cultures treated with nifedipine and the T-type channel blocker. We found that the inhibition of capacitative calcium entry resulted in inhibition similar to that achieved via L-type inhibition, while T-channel inhibition led to almost total blockade of the response. This result demonstrates that IL-6 release by ATP involves capacitative and L-type Ca^2+^ channels and confirms the clear dependence on T channels.

As in the case of the Ca^2+^ signal, it is likely that ATP-induced IL-6 secretion has a component dependent on Ca^2+^ release from the intracellular stores. The role of this component has been dissected using inhibitors of ryanodine and IP_3_ receptors (RyR and IP_3_R). To inhibit RyR, we used supra-micromolar concentrations of ryanodine and dantrolene, known blockers of the RyR receptor [41]. Figure 6A shows that both compounds impaired the initial part of the Ca^2+^ signal, as expected, for reduced release from intracellular stores. The IL-6 response to ATP was also inhibited by these compounds, as expected if Ca^2+^ release through RyR was involved in the stimulated secretion. To evaluate if cADPr, the endogenous agonist of the RyR, participates in IL-6 secretion, we treated cultures with an antagonist of this molecule, 8Br-cADPr, and with 78c, an inhibitor of ADP-ribosyl cyclase, the enzyme responsible for its synthesis. Both treatments induced a slight and statistically not significant reduction (Figure 5C), suggesting that cADPr is not involved in IL-6 secretion via ATP. This means that RyR does not need cADPr contribution to promote IL-6 secretion, suggesting direct activation by L-type channels.

Besides RyR, IP_3_R is the other main route for releasing Ca^2+^ from internal stores in response to agonists in skeletal muscle cells [17,41]. Therefore, we treated cultures with the compounds 2-APB and xestospongin C, two blockers of IP_3_R. This treatment induced a clear reduction in the peak phase of the Ca^2+^ signal, as can be observed in panels A and B of Figure 7. The result of the treatment was also a significant decrease in the IL-6 secreted in response to ATP stimulation (Figure 7C). The inhibitory effect of 2APB was stronger than the effect of xestospongin C on the IL-6 release, likely due to its greater inhibition of the Ca^2+^ signal.

If the two types of intracellular Ca^2+^ stores are activated by ATP and contribute to IL-6 release independently, their simultaneous inhibition should result in an additive reduction in IL-6 release. Therefore, we repeated experiments using 2-APB, ryanodine and a combination of both blockers. The simultaneous presence of ryanodine and 2-APB did not abrogate the IL-6 release, and there was no statistically significant difference between any of the inhibitors and their combination (Figure 7D).

To further substantiate the participation of calcium stores in IL-6 secretion, we used caffeine. This drug is known to release Ca^2+^ from RyR-bearing stores [42] and inhibits IP_3_R [43], and it has been reported to release IL-6 from skeletal muscle cells in culture [44]. Therefore, it can be expected that caffeine releases IL-6 in non-stimulated cultures and decreases the response to ATP, which is partially based on IP_3_R. We found here that application of caffeine 15 min prior to ATP challenge enhanced the resting IL-6 secretion; on the contrary, it reduced the ATP-evoked secretion, as expected by a compound interfering with the IP_3_R pathway (Figure 8).

Once established that the secretory response involves Ca^2+^ signalling elements, we assessed the participation of the Ca^2+^-binding protein calmodulin using its inhibitor calmidazolium. Figure 9A shows that the application of calmidazolium during the stimulation induced a statistically significant decrease in the ATP-induced IL-6 release compared to untreated cultures (*p* < 0.05). Given that calmodulin inhibition impairs the IL-6 response to ATP, we checked whether the inhibition of calcineurin and calmodulin-activated kinase II (CaMKII) also impaired the response. Treatment with the calcineurin inhibitor FK506 resulted in a reduction of ATP-evoked IL-6 release as compared to control cells (Figure 9B; *p* < 0.05). To rule out the possibility that the effect of FK506 was not mediated through calcineurin inhibition but by binding to FKBP proteins, known regulators of RyR, we used rapamycin, an FKBP-binding drug without effect on calcineurin, and found that it has no effect on IL-6 secretion. In the case of CaMKII, the use of a maximal concentration of its specific inhibitor KN93 only induced a slight reduction in the limit of significance compared to untreated cultures or to the inactive analogue of KN93, KN92 (Figure 9C). This indicates that the main signalling pathway downstream of calmodulin-mediating IL-6 secretion is not CaMKII but calcineurin.

Previous authors have shown that ATP stimulates IL-6 transcription in rodent muscle [11]. It is possible that the Ca^2+^ signals evoked by ATP operate at the transcriptional or post-transcriptional level. To test this question, we determined the changes in IL-6 mRNA levels in response to ATP. Figure 9D shows that ATP induced a clear increase that was not inhibited by the intracellular Ca^2+^ chelator BAPTA. In fact, IL-6 transcription was even enhanced in BAPTA-loaded cultures. This result indicates that the role of Ca^2+^ signals in the IL-6 secretion takes place at the post-transcriptional level. We also tested the effect of the T channel blocker NNC 35-0396 on the IL-6 expression. Similar to BAPTA loading, this treatment was without effect, as expected if T-type calcium channels operate only at the exocytosis process.

## 4. Discussion

Although there is extensive literature on the release of IL-6 and other myokines from skeletal muscle, the scarcity of our knowledge regarding the cellular mechanisms driving this endocrine secretion is surprising, especially considering the pathophysiological role of IL-6. Previous reports on the role of Ca^2+^ signals are fragmentary, and most of them have been conducted in rodent models. To our knowledge, there are only a few reports in human skeletal muscle, including IL-6 release in response to Ca^2+^ ionophore [31], RyR activation [28] and depolarization [45]. Our work reveals that Ca^2+^ influx via VOCC and Orai1, as well as Ca^2+^ release from RyR/IP_3_R-expressing Ca^2+^ compartments, mediate IL-6 release in response to ATP, a key cellular agonist in the developing skeletal muscle [37]. The L-type or Cav1.x Ca^2+^ channels are considered the voltage sensor element in the molecular mechanism coupling excitation of the sarcolemma to contraction [46] and to Ca^2+^ entry [47]. Cav1.x induces Ca^2+^ release from internal stores by interacting with RyR receptors of the sarcoplasmic reticulum [48]. This coupling is bidireccional because RyR can sensitize L channels to voltage changes [49]. Moreover, reports in adult and cultured rodent muscle have shown that this channel forms a multimolecular complex with other signalling molecules such as pannexin, caveolin and purinergic receptors [50,51].

In keeping with the above precedents, the present data indicate that both L channels and RyR receptors play a role in the stimulated release of IL-6. Although the study was not designed to investigate the regulation of the mechanisms, it is somehow surprising that treatment with antagonists for the cADPr route is without effect on IL-6 secretion because this messenger activates Ca^2+^ release from RyR. To explain the recruitment of RyR in the IL-6 secretion, a likely explanation would be a direct molecular interaction assembling L-channels and RyR during ATP stimulation, given the evidence described above.

The importance of VOCC channels for IL-6 secretion implies that depolarization is likely also involved. Our results prove that the secretion of IL-6 in skeletal muscle cultures relies on the entry of external Ca^2+^ ions, as shown by the effects of low Ca^2+^ medium, the non-specific blocker La^3+^, and the inhibitors of VOCC and Orai1 channels. Although it has been claimed that the entry of Ca^2+^ ions during excitation–contraction coupling in mouse adult muscle is vestigial [52], there is evidence for Ca^2+^ entry during activation [45]. As a whole, our data indicate that stimulated IL-6 secretion from cultured muscle cells requires proper Ca^2+^ signals driven by the entry of extracellular Ca^2+^ and by the release from RyR and IP_3_R Ca^2+^ stores. In muscle cells, ATP mobilizes intracellular calcium stores [53], and the presence of capacitative and store-operated Ca^2+^ channels is firmly established [54]. Though speculative, it seems possible that, once activated by the mobilization of IP_3_R-operated Ca^2+^ stores, Ca^2+^ influx via Orai channels might depolarize plasma membranes and trigger VOCC both directly and indirectly via the activation of cationic TRPC channels, which in turn would enhance depolarization [55,56]. This sequence would explain the strong inhibition in IL-6 release induced by the IP_3_R blocker 2APB. In addition, it is compatible with the lack of additive effect of inhibitors of IP_3_R and RyR observed in our experiments (Figure 10).

A remarkable finding is the fact that a specific blocker of the T-type Ca^2+^ channels achieved the deepest inhibition while showing no contribution to the whole-cell Ca^2+^ signal induced by ATP (Figure 4 and Figure 5). This result could be explained if T-channel operates as a requisite for the final secretion of IL-6 but not as a shaper of the whole-cell calcium signal (Figure 10). It is known that T-type Ca^2+^ channels contribute to neurotransmitter release in neurones and chromaffin cells [57]. Interestingly, there is indirect evidence linking IL-6 and T-channels. T-type inhibitors ameliorate the effects of inflammatory processes related to IL-6 [58,59], and IL-6 regulates T-type channels expression and function [60,61,62].

To date, most of the studies on the signal transduction for IL-6 secretion have been performed in rodent models and involve a number of intracellular signalling pathways as mediators for IL-6 release, such as Stat3 [11], JNK/MAPK [20,63], and AMPK [25,26]. These pathways would lead to the activation of the transcription factors NFKB, AP-1 [18] and Srf [24] that promote IL-6 secretion, while the epigenetic regulator HDAC5 [26] seems to inhibit it. Our data are in keeping with a previous report [17] showing that calmodulin and its downstream effector calcineurin mediate IL-6 secretion, while the calmodulin effector CaMKII has, if any, a residual role in this response.

Our data indicate that Ca^2+^ signals operate at the post-transcriptional level, in contrast to a previous report in rodent muscle [11]. The rationale for this discrepancy could be a difference either in the target for Ca^2+^ signals or in the positive feedback loop system reported by these authors. They proposed that the Ca^2+^ signal in response to ATP activates an early IL-6 transcription, which leads to further IL-6 secretion via the autocrine effect of released IL-6. In our experimental system, the initial IL-6 could be rather due to the activation of translation/exocytosis steps. This is in keeping with the strong effect of the specific T-channel inhibitor and the clear inhibitory effect of the translation blocker cycloheximide in IL-6 secretion [11] (and also unpublished results from our laboratory).

To our knowledge, previous reports on the participation of Ca^2+^ signals in IL-6 secretion are fragmentary and, in the case of human cells, almost inexistent. The evidence presented here describes for the first time in a comprehensive way the participation of the main elements of the Ca^2+^ signalling pathway in the stimulated IL-6 secretion from skeletal muscle cells. In addition to its previously reported participation in contraction, transcription and differentiation, this signal pathway is likely a physiological regulator of IL-6 secretion. Given the obvious importance of this myokine in a broad group of metabolic, immunological and pathophysiological processes, a deeper understanding of this topic deserves further investigation.

## 5. Conclusions

The present study shows that in cultured human skeletal muscle, the IL-6 secretion in response to ATP requires Ca^2+^ signals operating at the post-transcriptional level and involving both Ca^2+^ entry from the extracellular milieu and release from intracellular stores and the activity of downstream calmodulin/calcineurin pathway. Our data suggest that Ca^2+^ release from internal stores activates store-operated Orai1 Ca2+ channels, which in turn triggers the opening of voltage-operated Ca^2+^ channels.

## 6. Limitations

The conclusions of the study are limited to the stimulation with purinergic agonist ATP. Although physiological muscle contraction is due to depolarization and releases extracellular ATP, it is likely associated with other mechanisms leading to IL-6 release. In addition, the study does not take into account the temporal kinetics of the secretion, which could show a differential contribution of the Ca^2+^ signal elements.

## Figures and Tables

**Figure 1 biology-12-00968-f001:**
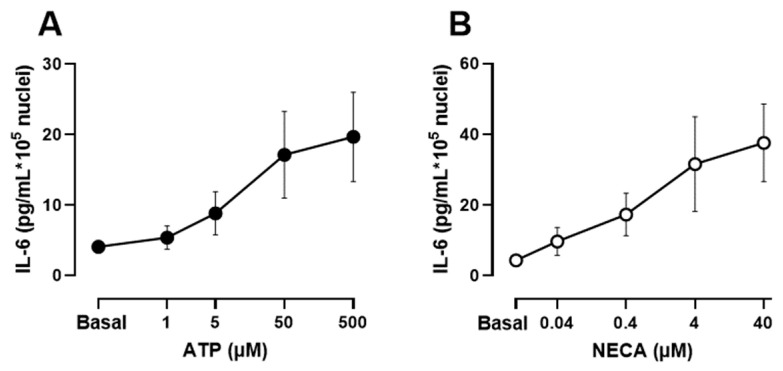
**IL-6 secretion in human skeletal muscle cells in response to increasing concentrations of ATPγS and the adenosine receptor agonist 5′-N-Ethylcarboxamidoadenosine (NECA)**. (**A**,**B**) Differentiated cultures were treated for 2 h in presence of the stimulus in serum-free medium. Data are mean ± s.e.m of 9 independent experiments from separate donors.

**Figure 2 biology-12-00968-f002:**
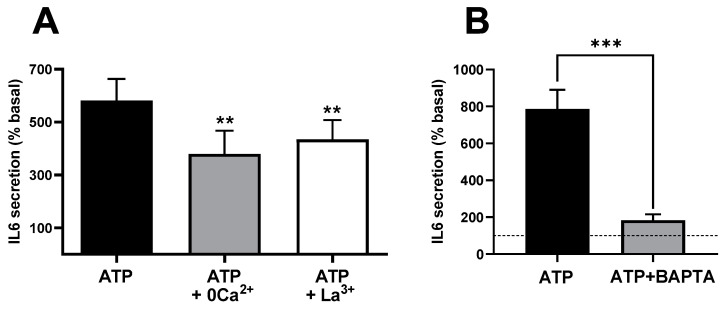
**Ca^2+^ signals and extracellular Ca^2+^ entry mediate IL-6 release in cultured human muscle cells**. (**A**) ATP-induced IL-6 secretion in the presence and absence of extracellular Ca^2+^ (0 Ca^2+^) or in the presence of 0.1 mM La^3+^. ** *p* < 0.01 vs. control ATP 500 µM (post-ANOVA; F = 21.18, *p* < 0.005) *n* = 10−5 (control and 0Ca^2+^ − La^3+^). (**B**) Effect of treating the culture with the intracellular Ca^2+^ chelator BAPTA-AM (50 µM, 20 min pre-treatment) in response to 500 µM ATPγS. *** *p* > 0.005 (paired *t*-test), *n* = 5. The dashed line represents the basal, non-stimulated secretion.

**Figure 3 biology-12-00968-f003:**
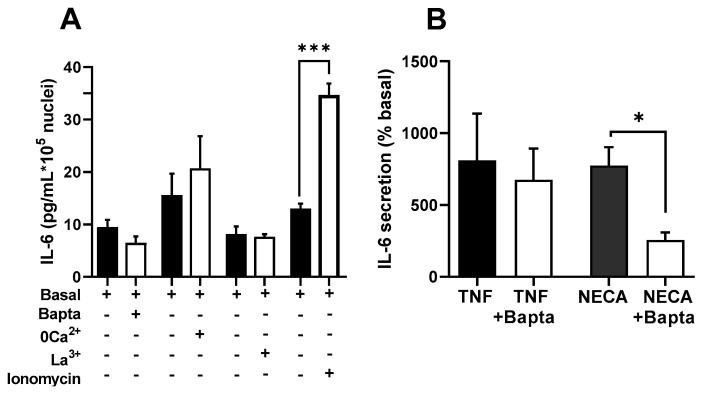
**Effects of BAPTA and inhibition of Ca^2+^ entry in IL-6 secretion in cultured human muscle cells are associated with Ca^2+^ signalling agonists.** (**A**) Resting secretion is not modified by treatment with Bapta-AM (50 µM) or with medium containing low Ca^2+^ or 0.1 mM La^3+^; *n* = 10 for Bapta, 5 for 0 Ca^2+^ and La^3+^. On the contrary, ionomycin (500 nM) induced a significant increase (*** *p* < 0.005, *t*-test, *n* = 7) (**B**) Treatment with Bapta-AM induced a significant reduction in response to NECA (40 µM), but not to TNFα (100 ng/mL). * *p* < 0.05, *t*-test, *n* = 5.

**Figure 4 biology-12-00968-f004:**
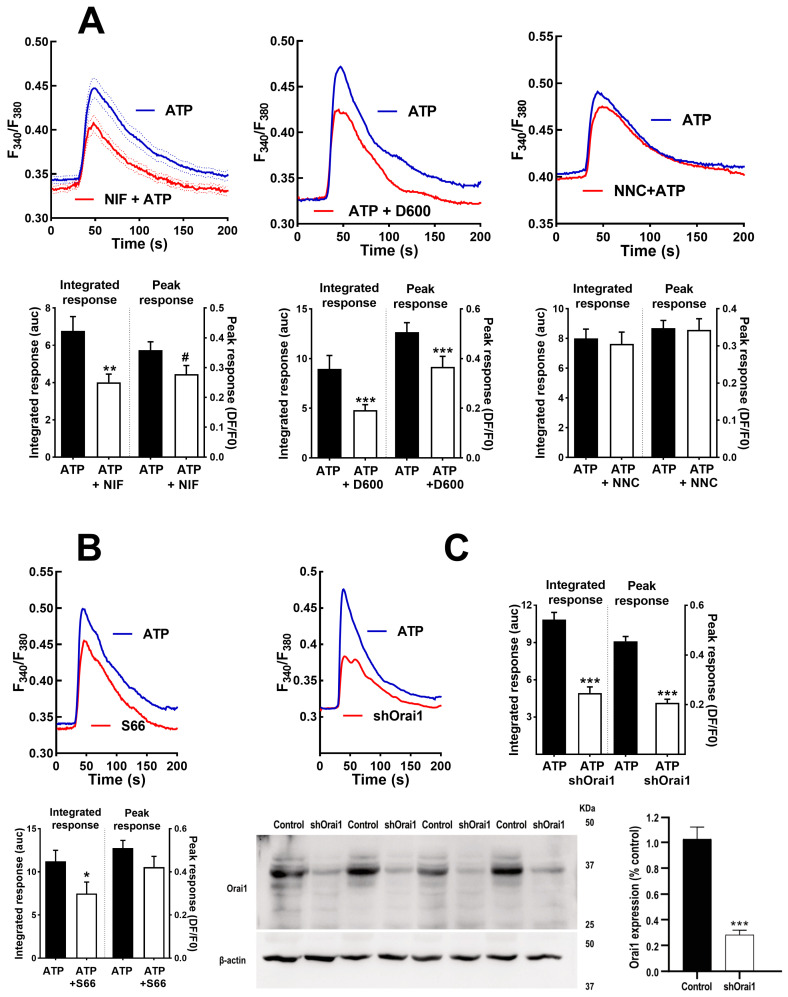
**Inhibition of voltage-operated Ca^2+^ channels and capacitative Ca^2+^ channels modifies Ca^2+^ signals in human skeletal muscle cultures.** (**A**) Average traces of the response to ATP in control cells (ATP) and cells pre-treated during 15 min with nifedipine (10 µM; NIF), verapamil (D600; 10 µM) or NNC55-0396 (1 µM; NNC). The histograms compare the sustained (Integrated response) and initial (Peak) phases of the responses. ** *p* < 0.01 *** *p* < 0.005 # *p* < 0.05 (one tailed) vs. control ATP, *t*-test. *n* = 91−63 for ATP-NIF, 39−32 for ATP-D600 and 76−28 cells for ATP-NNC. 5−8 independent experiments. (**B**,**C**) Average traces showing the effect of Synta 66 (1 µM, S66 15 min pre-treatment) and ablation of protein Orai1 (shOrai1) on the Ca^2+^ response to ATP. * *p* < 0.05 *** *p* < 0.005 vs. ATP alone, *t*-test. *n* = 58−38 cells for ATP-S66, 46−65 cells for ATP-shOrai. 4–8 experiments. The bottom panel in C shows Western blotting results of 4 separate samples of cultures transfected with the silencing plasmid shOrai1, including a comparison of Orai1 expression *** *p* < 0.005, *t*-test.

**Figure 5 biology-12-00968-f005:**
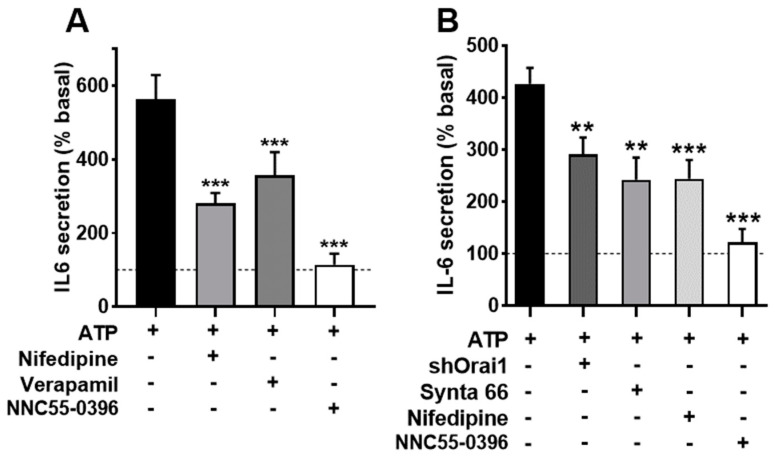
**Inhibition of voltage-operated Ca^2+^ channels and capacitative channels impairs IL-6 secretion in human skeletal muscle cultures.** (**A**) Cultures were stimulated with 500 µM ATPγS for 2 h in absence or presence of the inhibitors nifedipine (10 µM; NIF), verapamil (D600; 10 µM) or NNC55-0396 (1 µM). One-way ANOVA showed significative effect for the treatment (F = 12.5, *p* < 0.002); *** *p* < 0.005 vs. control ATP, multiple post-ANOVA comparison. *n* = 10 for control and nifedipine, and 5 for D600 and NNC. (**B**) Effect of shOrai1 transfection, Synta 66 (1 µM), nifedipine and NNC55-0396 on the ATP-evoked IL-6 release. One-way ANOVA F = 11.96, *p* < 0.0001; ** *p* < 0.01, *** *p* < 0.001 vs. control ATP, multiple post-ANOVA comparison *n* = 11 for control, 6 for shOrai and nifedipine and 5 for Synta66 and NNC55-0396.

**Figure 6 biology-12-00968-f006:**
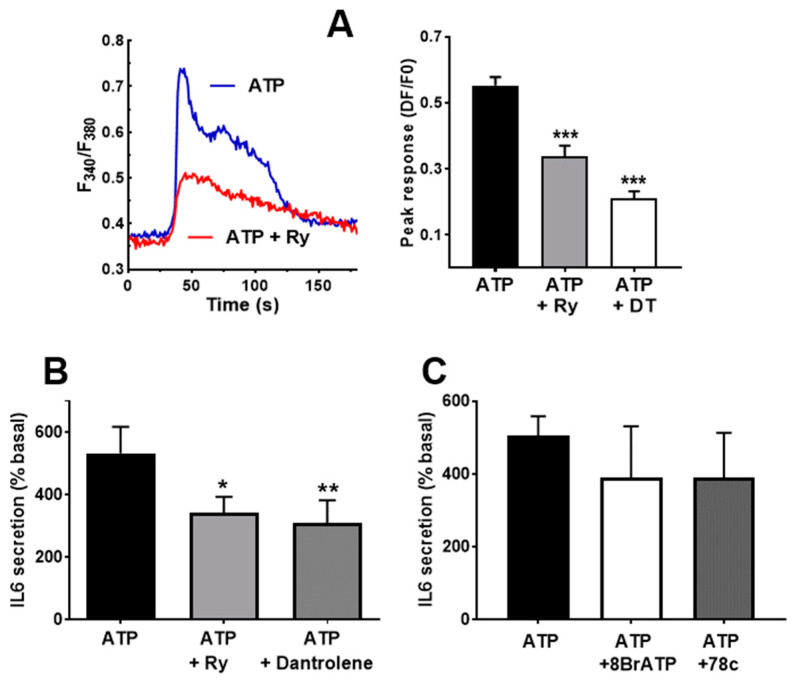
**Ryanodine receptor participate in IL-6 secretion**. (**A**) Representative traces showing the effect of Ryanodine (Ry) pre-treatment on the Ca^2+^ peak in response to ATP. The right-side panel represents the inhibition by ryanodine and dantrolene. One-way ANOVA F = 28.73, *p* < 0.0001; *** *p* < 0.005 vs. untreated, post-ANOVA test. *n* = 29−95 cells. (**B**) Effect of 10 µM ryanodine (Ry) and dantrolene (40 µM) in the IL-6 released in response to ATPγS (500 µM). * *p* < 0.05, ** *p* < 0.01 vs. control, post-ANOVA test (F = 12.23, *p* < 0.046 for treatment effect), *n* = 5. (**C**) Effects of 8BrcADPr (50 µM) and the CD38 inhibitor 78c (100 µM) in response to ATPγS. ANOVA analysis showed no effect for the treatment (*p* < 0.68), *n* = 4.

**Figure 7 biology-12-00968-f007:**
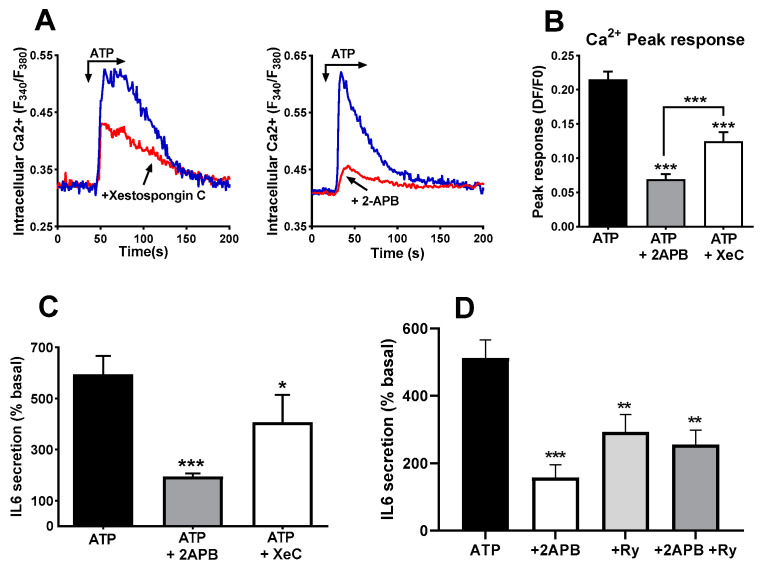
**Blockers of IP3 receptors impair ATP-induced IL-6 secretion**. (**A**) Representative record of ATP -induced Ca^2+^ signals in control cells and in cells pre-treated (15 min) with xestospongin C (4 µM) or 2-APB (50 µM). Panel (**B**) represents the comparison of the peak responses. ANOVA F = 41.25 *p* < 0.0001. *** *p* < 0.005 vs. control ATP, post-ANOVA comparisons or between 2APB and XC groups (*t*-test). *N* = 60 cells for 2APB, 70 for XeC and 110 for ATP alone. Eight experiments for control, five for 2APB and XeC. (**C**) Human skeletal muscle cultures were stimulated with ATPγS 500 µM in absence or presence of 2-APB (50 µM) or Xestospongin C (Xe C, 4 µM). * *p* < 0.05, *** *p* < 0.005 vs. control, multiple post-ANOVA comparisons (One-way ANOVA F = 11.6, *p* < 0.004 for treatment effect). *n* = 6 for 2-APB and 5 for xestospongin (**C**,**D**) IL-6 secretion in response to ATP alone or in the presence of 2-APB, ryanodine (Ry, 10 µM) or a combination of both. ANOVA F = 10.25, *p* < 0.001; ** *p* < 0.01, *** *p* < 0.005 vs. ATP alone, *n* = 4.

**Figure 8 biology-12-00968-f008:**
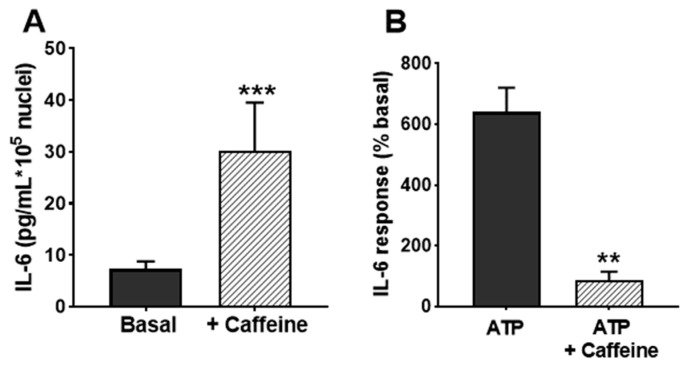
**Effects of caffeine on IL-6 release in cultured human skeletal muscle cells.** (**A**) IL-6 secretion in resting cultures in the absence and presence of 1 mM caffeine. (**B**) shows the effect of caffeine in response to 500 µM ATPγS (percentage respect to basal secretion). ** *p* < 0.01, *** *p* < 0.005 vs. control; *n* = 4.

**Figure 9 biology-12-00968-f009:**
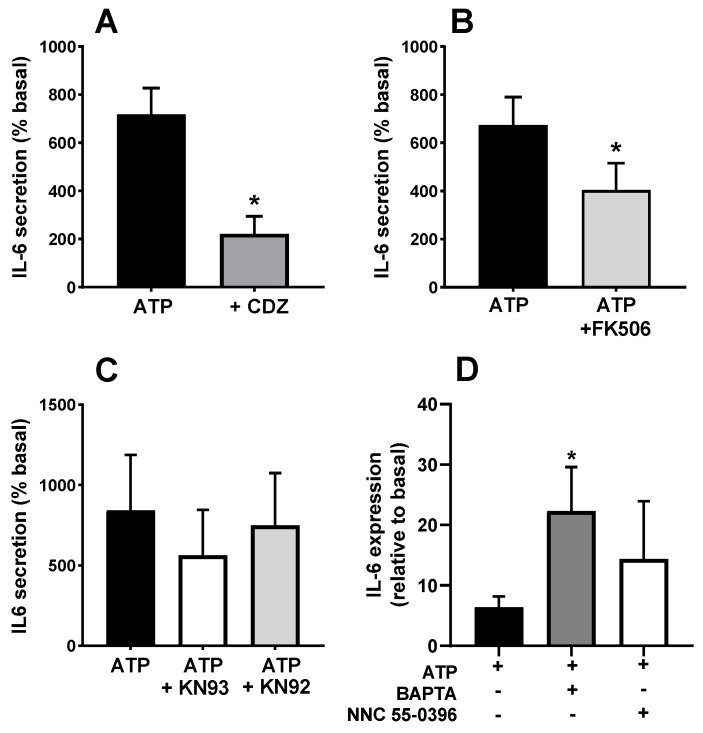
**Effects of calmodulin, calcineurin and translation inhibitors in the ATP-induced IL-6 secretion**. Human skeletal muscle cultures were stimulated with 500 µM ATPγS in the absence or presence of inhibitors of (**A**) calmodulin (calmidazolium, CDZ, 10 µM), (**B**) calcineurin (FK506, 10 µM) and (**C**) CaMKII (KN93 and its inactive analogue KN92, 3 µM). * *p* < 0.05 vs. ATP paired *t*-test *n* = 6 (**A**), 4 (**B**), 3 (**C**). ANOVA showed no difference for KN93 (F = 2.72, *p* > 0.2). (**D**) transcription of IL-6 (normalized to GAPDH) in response to ATP in the absence or presence of Bapta-AM (50 µM) or NNC 55-0396 (1 µM) * *p* < 0.05, *t*-test, *n* = 5.

**Figure 10 biology-12-00968-f010:**
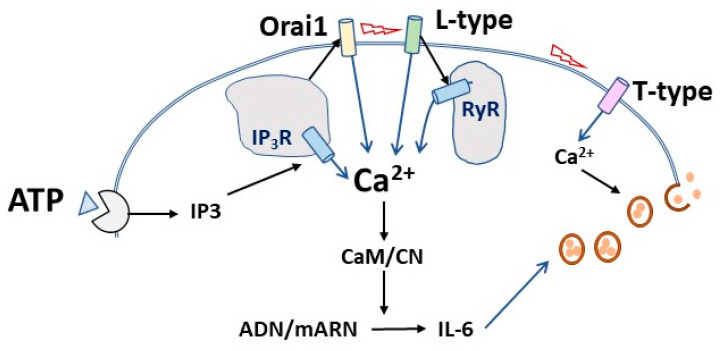
**Hypothesis for the sequential events in the calcium-mediated IL-6 release induced by ATP in cultured muscle cells**. Upon ATP binding to purinergic receptors, IP3 synthesis leads to Ca^2+^ release from intracellular stores, thereby activating Ca^2+^ entry through capacitative Orai1 channels. This could elicit membrane depolarization and activation of L-type calcium channels, resulting in further Ca^2+^ increase through direct influx and direct activation of the associated ryanodine receptors (RyR) of intracellular stores. The Ca^2+^ signal activates IL-6 translation that would be released by the operation of voltage-activated T-type calcium channels.

## Data Availability

Data are available upon request to the corresponding author.

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
