# Peer review of "Secretion of Interleukin 6 in Human Skeletal Muscle Cultures Depends on Ca2+ Signalling"

_biology, 2023, doi:10.3390/biology12070968_

Round 1

Reviewer 1 Report (Previous Reviewer 1)

Calle-Ciborro and colleagues resubmitted their work after the first round of revision providing new important data by performing genetic ablation of some of the Ca2+ handling components that participate to the Ca2+-dependent IL6 secretion (i.e. Orai1), by carrying out Ca2+ measurements to assess the effect of treatment on the intracellular cation dynamics and by investigating at which level ( protein translation vs protein secretion)  the regulation of IL6 production occurs. I recognized the effort of the authors to respond to some of my major points and appreciated the new findings which undoubtedly added value to the work.

Nevertheless, half of my major points were not considered (point #2,5,6 and 7). Thus, I have found several unaddressed issues in the revised manuscript (see below), which I encourage the authors to take into account, and solve, before considering the article suitable for publication.

Major issues:

1)      (see my previous major point #5) I have a concern about the conclusion that Ca2+ is not involved in basal IL6 production drawn from the data presented in fig.3.  Is the Elisa test to measure IL6 secretion sensitive enough to detect variations in the unstimulated (basal) IL6 release? From Fig. 1 graphs it appears that basal IL6 production is very close to 0 pg/mlx10^5 nuclei (see NECA panel). I am wondering if the assay might be able to detect IL6 levels lower than that. Could the authors provide an experimental confirmation of the sensitivity of the test in this range (by inhibiting basal IL6 secretion for example?)

2)      Fig. 3A the X-axis legend could be reduced by deleting the line of “Basal”, which is indeed useless. Moreover, it apparently shows three identical conditions, which represent the basal IL6 secretion in un-treated cells (i.e. first bar-black, fourth bar-white and fifth bar-black), having definitively different values. How could this happen?

3)      Fig. 4 shows that L-type channel blockers inhibited ATP-induced intracellular Ca2+ increase, while NNC T-type channel blocker did not. However, the effect of NNC on inhibiting the ATP-induced IL6 release is much stronger compared to the other compounds. The authors did not comment on this finding. I would like to have an explanation for this apparently contrasting finding.

4)      The authors mentioned the reduction of ER store Ca2+ refilling as a possible explanation for the “expected” differences in the Ca2+ peak values of Synta66 vs shOrai1 treated cells (Fig. 4 B and C), however no experimental data are provided. The authors should quantify the ER Ca2+ content in the two conditions in order to verify their hypothesis.

5)      (see my previous major point #7) The negative effect of RyR and IP3R blockers seem perfectly complementary in terms of ability to inhibit IL6 release. Do the authors thought of trying experiments with the two inhibitors together, in order to test if the combination is able to completely abrogate the IL6 secretion? It would be interesting to have the result of this experiment.

6)       The authors only applied a negative regulation approach by inhibiting/downregulating different molecular players to demonstrate their role in the ATP-induced IL6 secretion in muscle cells. They did not try to promote / enhance IL6 secretion by stimulating the cytosolic Ca2+ mobilization (by Orai1 or IP3R agonist for example). This approach would be crucial to demonstrate that intracellular Ca2+ increase is not only necessary to the ATP-mediated release of IL6, but also able to positively modulate it.

7)      (see my previous major point #2) I appreciate the new experiments about the inhibition of protein translation using cycloheximide. However, this approach did not answer, or answered only partially, to the question whether the regulation of IL6 production occurs at the transcriptional or post-transcriptional level. Indeed, I asked for the assessment of IL6 gene expression, which was not considered in the new version of the article. On the other hand, the results of the cycloheximide experiments rose an additional comment. Since the ATP-induced release of IL6 is measured at 2h after the stimulus addiction, I am wondering if it is feasible that the effect of cycloheximide on the de novo IL6 protein synthesis may be revealed in such a short time window. This limited time frame leads me to speculate about a rapid translation of mRNA molecules already accumulated and ready to be translated. Once more, the quantification of IL6 messenger levels would be crucial to shed light on this issue.

Minor points:

·       Figure legends of figure 1 and 2 are interspersed within the text, many lines far from the Figure itself. It will be more helpful for the reader if legends would be placed just under the image panels so be presented as a whole with the Figure, as it occurs for the other figures of the manuscript.

·       I suggest modifying the phrase “in the presence of 0 Ca2+” by replacing with the more appropriated sentence: “in the absence of extracellular Ca2+”, please correct throughout the manuscript.

·       At line 247, legend of Fig.2: “A) ATP-induced IL6 secretion…”, ATP-induced should be added.

·       At line 263, TNFa has to be corrected and written with Greek letter.

·       At line 300, Ca2+ superscript should be used

·       At line 309, please correct as follows: “this channel is absolutely required…”

·       A representative WB experiment is sufficient to show the efficacy of shOrai1 in Fig. 4. The quantification of WB bands in the last graph of the figure is clearly confirming the consistency of the downregulation.

·       Remove “specific” at Fig.4 legend line 354.

·       At line 364, replace “can be dissected” with “has been dissected”

·       At line 373, replace “statistical” with “statistically”.

·       At line 375, correct the sentence to: “RyRs do not need cADPr contribution to promote IL6 secretion …”

·       I encourage the authors to uniform the symbol and acronyms in Fig. 6 (Dantrolene is written once extended and once as DT)

·       The Friedman statistical test used in Fig.6 is appropriated for comparison of repetitive measures on the same samples after different treatments, which is not the case (since the IL6 measures in the paper derive from independent cell samples, if I get it correctly). Could the authors justify the use of this test for their analysis?

·       At line 438, “posterior” is absolutely not appropriate, use instead “post-translational”

·       At line 440, I suggest to rephrase as follows: “..stimulates IL6 secretion mainly promoting IL6 translation”

No major issues, I only suggested some minor modifications, see minor comments in the previous report form.

Author Response

Reviewer 2 Report (Previous Reviewer 2)

lle-Ciborro et al assessed the Ca dependency for Il-6 sectretion from skeletal muscle cells and found that IL-6 secretion from human primary skeletal muscle cells is enhanced by several fold upon increase in the cytosolic Ca2+ level as a results of Ca2+ influx and release from cytosolic store. The study has validated this finding using solely a molecular and pharmacological approach.  The study is well written and easy to read and statistical analysis are well chosen. The authors has addressed all the issues arised in my previous report appropriately. 

This paper may be accepted for publication in Biology

minor english correction required 

Author Response

Many thanks for reviewing our manuscript

Reviewer 3 Report (Previous Reviewer 3)

The authors addressed my comments satisfactorily

Author Response

Thank you very much for revising our manuscript

Round 2

Reviewer 1 Report (Previous Reviewer 1)

Calle-Ciborro and colleagues resubmitted a revised version of their manuscript considering all the comments I pointed out in my report, adding new experimental evidence and modifying the text accordingly. I am quite satisfy of their point-by-point reply and find the paper much improved after the revision. I think is is now acceptable for publication in the Biology Journal.

However, some minor errors still need to be corrected in the text and I will list theme here below:

1)      Add a space between the legend of fig. 2 and the remaining text (line 243).

2)      The level of statistical significance should be indicated by the asterisks at line 280: *** p < 0.005 (asterisks are missing)

3)      The first line of Legend of fig. 6 should be moved below the figure (line 360)

4)      In the legend of X-axis of Panel D of fig. 7 2-APB is mis-spelled.

5)      Line 433: do the authors mean post-transcriptional rather than post-translational?

6)      Line 509: here is the opposite: do the authors mean translation rather than transcription?

Author Response

Dear revisor,

thank you very much for reviewing our manuscript.

We have corrected all the errors as indicated in your report. In the case of Points 5 and 6, you are right, translation and transcription were wrongly exchanged.

This manuscript is a resubmission of an earlier submission. The following is a list of the peer review reports and author responses from that submission.

Round 1

Reviewer 2 Report

Calle-Ciborro et al assessed the Ca dependency for Il-6 sectretion from skeletal muscle cells and found that IL-6 secretion from human primary skeletal muscle cells is enhanced several fold upon increase in the cytosolic Ca2+ level as a results of Ca2+ influx and release from cytosolic store. The study has validated this finding using solely a pharmacological approach.  The study is well written and easy to read however, I have several concerns that need to be addressed before final acceptance

1)      The introduction the authors cited several studies that have investigated the role of Ca2+ signalling on IL-6 secretion from muscle of human and rodents and mouse lymphocyte, and the only gap left is that the Ca2+ influx pathways have not been studied, why then to study the Ca2+ release from intracecullar store, please define the aim in a better way.

2)      The first method section, the authors need to modify it as isolation and culturing of human primary skeletal muscle cells.

3)      Why and to what you differentiated the cells?

4)      It is appreciated that the authors listed the reagent and drugs used as separate section however, it is not described in the methodology where most of these drugs were used and the concentration of these drugs. The authors need to address this issue by describing were they were used and concentration.

5)      It is important to mention the solvents in which these drugs were dissolved.

6)      Line 83> several day, how many days exactly?

7)      In figure 1 both ATPγS and (NECA) increased IL6 secretion, but where is the control where you cultured and measured IL-6 release in the presence of the vehicle only

8)       In figure 2 the control is absent also to ATP as well as BAPTA and in Fig 2b the authors claimed 0Ca2+ upon incubation with 1 mM EGTA this is a stretch, I suggest to replace 0 Ca2+ with 1 mM EGTA

9)      Please pay attention to the statistics information described in the legend of each figure, mention the test type, replicate numbers, post hoc test, significance level etc. and in figure 2 why paired t test was used? How about those data presented in figure 7 and 8?

10)   Please correct several typos throughout the manuscripts, cytoquine (line 47), write  Ca2+ not Ca2+, during several days to be change into for several days (line 83), among others.

Reviewer 3 Report

The manuscript is interesting and has merit. I have a few comments:

1.     Introduction section does not adequately describe the current understanding of IL-6 realease from skeletal muscle cells. Please expand the introduction.

2.     Which muscle were primary cultures prepared from?

3.     Cells from how many donors were used, how old were the donors, were they healthy before the cell harvest, did they take any drugs that could influence skeletal muscles? These data should be reported in the manuscript.

4.     Were experiments done on myoblasts, myotubes? How pure was the myoblast culture (myoblast to fibroblast ratio)?

5.     Please add also a limitation section.

6.     A figure of known molecular mechanisms in IL-6 release with emphasis on a Ca2+ signal part would be beneficial.

7.  Do you think the role of Ca2+ is the same in animal muscle cells?
